# The Lacto-Tetrapeptide Gly–Thr–Trp–Tyr, β-Lactolin, Improves Spatial Memory Functions via Dopamine Release and D1 Receptor Activation in the Hippocampus

**DOI:** 10.3390/nu11102469

**Published:** 2019-10-15

**Authors:** Tatsuhiro Ayabe, Yasuhisa Ano, Rena Ohya, Shiho Kitaoka, Tomoyuki Furuyashiki

**Affiliations:** 1Research Laboratories for Health Science & Food Technologies, Kirin Holdings Company Ltd., 1-13-5 Fukuura Kanazawa-ku, Yokohama-shi, Kanagawa 236-0004, Japan; Yasuhisa_Ano@kirin.co.jp (Y.A.); Rena_Ohya@kirin.co.jp (R.O.); 2Division of Pharmacology, Kobe University Graduate School of Medicine, Kobe 650-0017, Japan; skitaoka@med.kobe-u.ac.jp (S.K.); tfuruya@med.kobe-u.ac.jp (T.F.); 3AMED-CREST, Chiyoda-ku, Tokyo 100-0004, Japan

**Keywords:** β-lactolin, dopamine, dopamine D1 receptor, spatial memory, whey

## Abstract

Scope: Peptides containing tryptophan–tyrosine sequences, including the lacto-tetrapeptide glycine–threonine–tryptophan–tyrosine (GTWY) and β-lactolin, from β-lactoglobulin in whey enzymatic digestion, enhance hippocampus-dependent memory functions, which are blocked by the systemic administration of dopamine D1-like antagonist. In this study, we investigated the role of the hippocampal dopaminergic system in the memory-enhancing effect of β-lactolin. Methods and Results: The results of in vivo microdialysis revealed that oral administration of β-lactolin increased the extracellular concentration of dopamine in the hippocampus and enhanced both spatial working memory, as measured in the Y-maze test, and spatial reference memory, as measured in the novel object location test. These memory-enhancing effects of β-lactolin, but not the baseline memory functions, were impaired by the knockdown of the dopamine D1 receptor subtype in the hippocampus. β-Lactolin also enhanced object memory, as measured by the novel object recognition test. However, D1 knockdown in the hippocampus spared this memory function either with or without the administration of β-lactolin. Conclusions: The present results indicate that oral administration of β-lactolin increases dopamine release and D1 receptor signaling in the hippocampus, thereby enhancing spatial memory, but it may improve object memory via a separate mechanism.

## 1. Introduction

As the size of the aged population is rapidly growing worldwide, cognitive decline and dementia have become serious health problems. Given the lack of post-onset therapeutic strategies for dementia, prevention through changes in daily lifestyle, such as dietary habits and exercise, has received increasing attention. Several epidemiological studies have suggested that consuming fermented dairy products, such as cheese and yogurt, reduces the risk of cognitive decline and dementia [1,2,3]. In a previous study of the underlying mechanisms of these findings, we identified the preventive effects of a *Penicillium candidum* fermented dairy product (i.e., Camembert cheese) against Alzheimer’s disease (AD) in a mouse model of AD [4,5].

In the process of identifying the ingredients responsible for our previous findings, we screened bioactive peptides from digested β-casein and found KEMPFPKYPVEP peptides, which improve memory function in mice [6]. We further determined the tryptophan–tyrosine (WY) sequence-containing peptides that were abundant in Camembert cheese, including a lacto-tetrapeptide of glycine–threonine–tryptophan–tyrosine (GTWY) and β-lactolin, from β-lactoglobulin in whey digestion, and found that administrating these peptides improved hippocampus-dependent memory functions in mice [7]. Investigating the structure–activity relationship, certain dipeptides containing N-terminal tryptophan, represented in WY peptide, improved those hippocampus-dependent memory functions [8]. Pharmacokinetic study has shown that β-lactolin is effectively distributed to brain areas, including the hippocampus. In vitro, β-lactolin inhibited monoamine oxidase (MAO)-B activity, which, is involved in the metabolism of dopamine, and increased the dopamine content in the hippocampus. Moreover, systemic administration of SCH23390, a D1-like receptor antagonist, attenuated the memory-enhancing effects of β-lactolin. These findings led us to hypothesize that hippocampal dopaminergic signaling is involved in the memory-enhancing effects of β-lactolin. However, there is, as yet, no direct evidence for this notion, and the question as to how the dopaminergic system is involved in the memory-enhancing effects of β-lactolin remains unanswered.

Dopamine exhibits its functions through multiple receptor subtypes, specifically D1-like (D1 and D5) and D2-like (D2, D3, and D4) receptors, and previous reports have shown that dopamine D1-like receptor activation enhances hippocampus-dependent memory functions [9,10,11,12]. Despite D1-like receptors being expressed in several brain areas, including the hippocampus, most studies have not identified the specific site where D1-like receptors exert their memory-enhancing functions. In addition, as selective pharmacological drugs are lacking, most studies have failed to discriminate between dopamine receptor subtypes. Recently, brain region-specific knockdown of the dopamine D1 receptor subtype was achieved through injection of adeno-associated viral vectors expressing artificial microRNA (miRNA) that target the D1 receptor subtype [13]. This technique will help to clarify the roles and actions of the dopaminergic system in the memory-enhancing effects of β-lactolin.

In the present study, using in vivo microdialysis, we found that β-lactolin increased the extracellular concentration of dopamine in the hippocampus. We also found that knockdown of the dopamine D1 receptor subtype in the hippocampus abolishes the memory-enhancing effects of this peptide on spatial, but not object, memory functions. These results indicate the hippocampal D1-dependent and independent mechanisms of action of β-lactolin.

## 2. Experimental Section

### 2.1. Materials

GTWY peptide (β-lactolin; purity: 98%) was purchased from NARD Chemicals, Ltd. (Amagasaki, Japan). Dopamine and scopolamine were purchased from Sigma Aldrich Co. (St. Louis, MO, USA).

### 2.2. Animals

Male, 7-week-old, ICR mice were purchased from Charles River Japan Inc. (Tokyo, Japan). The animals were kept at room temperature (23 °C ± 1 °C) under a constant 12-h light/dark cycle (light period from 8:00 a.m. to 8:00 p.m.) and were fed a standard rodent diet (CE-2 (Clea Japan, Tokyo, Japan)). Behavioral pharmacological tests were conducted in a sound-isolated room. All the animal care and experimental procedures were performed according to the guidelines of the Animal Experiment Committee of the Kirin Holdings Company Ltd., and all efforts were made to minimize suffering. The Animal Experiment Committee of Kirin Holdings Company Ltd. approved all the studies; the approval IDs were AN10529-Z00 and AN10599-Z00.

### 2.3. In Vivo Microdialysis

In vivo microdialysis was performed as described previously [14]. In brief, each animal was placed in a stereotaxic frame (Narishige, Tokyo, Japan) under pentobarbital (Somnopentyl; Kyoritsu, Tokyo, Japan) anesthesia, and then an incision was performed on the skin covering the skull under local anesthesia with lidocaine (Xylocaine Jelly; AstraZeneca, London, UK). Next, a hole was drilled through the skull, and a guide cannula (Eicom) was inserted into the hippocampal region (bregma: −3.5 mm anterior, −3.0 mm lateral, and 1.8 mm vertical). A dummy cannula was inserted, and animals were allowed for at least 5 days to recover. Then, a microdialysis probe (Eicom) was inserted into the hippocampal region through the guide cannula, extending 1 mm beyond its tip. This probe was perfused with Ringer’s solution which was described previously [14] at a flow rate of 1.0 μL/min. From 4 h after the probe was inserted, the dialysates were collected every 20 min. The dopamine concentration was quantified using high-performance liquid chromatography, coupled with electrochemical detection (HPLC-ECD; Eicom, Kyoto, Japan). Following dialysate collection for at least an hour as the baseline, β-lactolin was administered orally.

### 2.4. Quantitative Reverse Transcription Polymerase Chain Reaction (RT-PCR)

The levels of mRNA in the hippocampus were measured using quantitative RT-PCR. Mice expressing either control miRNA or D1 miRNA in the hippocampus were decapitated to enable the removal of the hippocampus, and was homogenized. The total RNA was purified using an RNeasy Mini Kit (QIAGEN, Hilden, Germany) and was then reverse-transcribed (2.5 μg) using a High Capacity cDNA Reverse Transcription Kit (Applied Biosystems, Foster City, CA, USA). The relative RNA levels were quantified using SYBR Green Real-Time PCR Technology (Takara Bio Inc., Shiga, Japan). The data were then normalized for glyceraldehyde-3-phosphate dehydrogenase values. The primers that were used for PCR are described in Table 1.

### 2.5. Adeno-Associated Virus (AAV) Injection

The procedures used in this study for AAV injection to knock down the dopamine D1 receptor were performed, as described previously [13]. This study used either AAV-expressing artificial miRNA targeting the dopamine D1 receptor (AAV10-EF1α-DIO-EmGFP-D1 miR) or control miRNA (AAV10-EF1α-DIO-EmGFP-ctrl), with Emerald Green Fluorescent Protein (EmGFP) under the EF1α promoter only in the presence of Cre recombinase (AAV10-EF1α-DIO-EmGFP-D1 miR) and expressing control miRNA (AAV10-EF1α-DIO-EmGFP-ctrl). These AAVs were mixed with another AAV-expressing Cre under the CMV promoter (AAV10-CMV-Cre). Eight-week-old ICR mice were anesthetized by pentobarbital and a glass capillary was inserted with the same procedure of in vivo microdialysis, and the AAV solutions were injected into both dorsal and ventral CA1 regions of the hippocampus (from bregma: posterior, −3.5 mm; lateral, −3 mm; ventral, −3.8 mm and −1.8 mm). The animals were allowed at least 3 weeks to recover prior to behavioral tests. After the behavioral experiments, the animals were sacrificed, and coronal brain sections (10 μm) were cut using a cryostat (CM3050 S; Leica Biosystems, Nussloch, Germany). The location of the AAV injection in one hemisphere was confirmed using emGFP fluorescence, and animals that did not express hippocampal fluorescence were excluded from further analyzes. RT-PCR was used to quantify hippocampal D1 mRNA in the other hemisphere. Within the D1 miRNA group, mice retaining D1 mRNA over 90% of the control miRNA group were excluded from further analysis.

### 2.6. Y-Maze Test

Spatial working memory was tested using the Y-maze test, as described previously [7]. The Y-maze comprised three black arms positioned at equal angles, each arm measuring 250 mm in length and 50 mm in width and having a 200-mm-high wall. The tests were observed through a digital video camera mounted on the ceiling. Either β-lactolin or vehicle (distilled water) were administered orally 60 min prior to the test. Each animal was placed in the Y-maze at the end of an arm and allowed to explore for 8 min. Spontaneous alternation behavior was defined as the mouse entering each of the three arms successively in overlapping triplet sets. The formula used to calculate spontaneous alternation (%) was as follows: [(number of spontaneous alternation behaviors)/(total number of arm entries −2)] × 100.

### 2.7. Novel Object Recognition Test (NORT)

Object recognition memory was measured using the NORT, as described previously [7]. In brief, the experimental apparatus used in this study was a square open field (40 cm in length, 40 cm in width, and 40 cm in height) made of gray polyvinyl chloride. The objects comprised two pairs of wooden blocks. Each object was placed in a corner, on the same side. The test consisted of a period of acquisition and a period of recall. The animals were acclimated to the experimental room for at least 16 h before the test. Then, each mouse was placed into the square open field and allowed to explore freely for 10 min. The recall was conducted after 24 h by replacing one of the objects in each pair with a novel object (a wooden white sphere). Each mouse was placed in the open field for 5 min, and the time spent exploring both the familiar and the novel objects was measured. Either β-lactolin or vehicle (distilled water) was administered orally 60 min before both the acquisition period and the recall period. The discrimination index was calculated using the following formula: (novel object exploration time-familiar object exploration time)/(total exploration time).

### 2.8. Novel Object Location Test (NOLT)

The NOLT was performed as described previously [15]. In brief, visual cues (black, white, or striped-pattern pictures) were placed on the wall of the square open field, so that the mice distinguish each direction. The objects used were brown glass vials. After acclimation to the experimental room, each mouse was placed in the open field for 10 min, without any objects, after which it was returned to the home cage. After 24 h, the acquisition was conducted by placing each mouse in the apparatus for 5 min, with two objects placed at the corners of the same side. After 4 h, the recall test was conducted by reintroducing each mouse to the apparatus for 8 min, with two objects placed at the two diagonal corners. Either β-lactolin or vehicle (distilled water) was administered orally 60 min before both the acquisition period and the recall period. Similar to the NORT, the time the animals spent exploring both the familiar object and the novel object was measured, and the discrimination index was calculated.

### 2.9. Dosage Information

In all animal experiments, β-lactolin was administered at a dosage of 1 mg/kg by oral gavage. In this study, the dosage of β-lactolin was determined according to our previous report [7]. The human equivalent dosage of β-lactolin is 0.08 mg/kg, calculated by multiplying the human equivalent dosage modulus (0.08). Either water or β-lactolin was administered 60 min prior to the Y-maze test. In the NORT and the NOLT, either water or β-lactolin was administered 60 min prior to both the acquisition period and the recall period.

### 2.10. Statistical Analysis

Results are presented as the mean ± standard error of the mean (SEM). Two-group comparisons were analyzed using the Student’s *t*-test. Values at respective time points with in vivo microdialysis were analyzed using a repeated-measures two-way analysis of variance (ANOVA), followed by Bonferroni’s test. All other experimental data were analyzed using one-way ANOVA, followed by the Tukey–Kramer test. All other statistical analyzes were conducted using the Ekuseru–Toukei 2012 software program (Social Survey Research Information, Tokyo, Japan). A level of *p* < 0.05 was considered statistically significant.

## 3. Results

### 3.1. β-Lactolin Increases the Extracellular Concentration of Dopamine in the Hippocampus

The effects of β-lactolin on the extracellular dopamine concentration in the hippocampus were studied using in vivo microdialysis. A significant interaction of time × treatment was observed (F10,190 = 2.4812, *p* = 0.008). Tests of simple effects revealed that single oral administration of β-lactolin increased extracellular dopamine levels significantly compared with the baseline (F10,190 = 4.5814, *p* < 0.001), but there was no significant difference observed in those of the control group (F10,190 = 0.4370, *p* = 0.9269; Figure 1). Comparisons of each time period revealed that the mice treated with β-lactolin had a significantly higher level of extracellular dopamine than did the control mice. These findings suggest that oral administration of β-lactolin increases the extracellular concentration of dopamine in the hippocampus.

### 3.2. β-Lactolin Improves Spatial Working Memory via the Dopamine D1 Receptor in the Hippocampus

We have found previously that systemic treatment with SCH23390, a D1-like antagonist, eliminated the enhancing effect of β-lactolin on spatial working memory in a scopolamine-induced amnesia mouse model. This receptor was knocked down in the hippocampus to examine the role of the hippocampal dopamine D1 receptor subtype in hippocampus-dependent memory functions. To this end, artificial miRNA targeting either the D1 receptor or control miRNA was expressed in the hippocampus. The findings revealed a significant decrease in the mRNA level of the D1 receptor in the D1 miRNA group compared with the control miRNA group, whereas the levels in the other dopamine receptor subtypes were not affected (Appendix A). As we have reported previously, this method of D1 receptor knockdown also reduces the protein levels of the D1 receptor [13]. Using the Y-maze test, we studied the effects of β-lactolin on spatial working memory. As reported previously, β-lactolin increased spontaneous alternation in the control miRNA group [7]. This enhancement was eliminated by knockdown of the D1 receptor in the hippocampus (F3,21 = 4.496, *p* = 0.0138, Figure 2a). There was no significant difference in the number of arm entries between each group (F3,21 = 0.396, *p* = 0.7571, Figure 2b). These findings suggest that the dopamine D1 receptor subtype in the hippocampus is vital to the effect of β-lactolin in enhancing spatial working memory, as measured in the Y-maze test.

### 3.3. The Dopamine D1 Receptor in the Hippocampus Is Vital for the Enhancing Effect of β-Lactolin on Spatial Reference Memory

Previously, we found that β-lactolin enhances object recognition memory in the NORT [7]. Here we investigated whether the dopamine D1 receptor subtype is involved in this effect. Consistent with our previous reports, β-lactolin increased significantly both the time spent exploring a novel object and the discrimination index in the control miRNA group (F3,24 = 6.473, *p* = 0.0023, Figure 3a,b). The time spent exploring a novel object and the discrimination index was also increased in the D1 knockdown mice by administration of β-lactolin, though there was no significant change. The total exploration time was not affected by any treatment (F3,24 = 1.265, *p* = 0.3086, Figure 3c). These findings suggest partial involvement of the dopamine D1 receptor subtype in the hippocampus, though it is not vital for the effect of β-lactolin on enhancing object memory.

This result prompted us to speculate that the hippocampal D1 receptor may contribute to spatial memory, rather than object memory. To investigate this, we carried out the NOLT, which is designed to test spatial reference memory of object locations by changing the location of one of two objects in similar behavioral chambers. Oral administration of β-lactolin increased both the time spent exploring a novel location and the difference between the proportions of the time spent exploring a novel location and a familiar location (i.e., the discrimination index) (F3,23 = 8.652, *p* < 0.001, Figure 4a,b). Knockdown of the hippocampal D1 receptor subtype eliminated this enhancement. The total exploration time was not affected by any treatment (F3,23 = 0.3633, *p* = 0.7801, Figure 4c). These findings suggest that the dopamine D1 receptor in the hippocampus is vital to the effect of β-lactolin in improving spatial reference memory, whereas this peptide may improve object recognition memory via a distinct mechanism.

## 4. Discussion

As we have reported previously, peptides containing the WY sequence, such as the GTWY peptide, β-lactolin, from whey enzymatic digestion, enhance hippocampus-dependent memory functions. However, the question of how the hippocampal dopaminergic system is involved in this memory-enhancing effect remains largely unanswered. In this study, we found that oral administration of β-lactolin increased the extracellular concentration of dopamine in the hippocampus. Although β-lactolin administration enhanced both spatial and object memory functions, the dopamine D1 receptor in the hippocampus appears to be involved in a different way. Knockdown of the dopamine D1 receptor in the hippocampus eliminated the effect of β-lactolin on both spatial working memory and reference memory, as measured in the Y-maze and the NOLT, respectively. Conversely, D1 knockdown did not affect the β-lactolin-induced improvement of object recognition memory. These findings suggest that β-lactolin enhances spatial memory functions through the hippocampal dopamine D1 receptor, whereas it enhances object memory via a separate mechanism.

We found previously that β-lactolin directly inhibits the activity of MAO-B in vitro. As β-lactolin is effectively distributed in the hippocampus 1 h after oral administration, MAO-B inhibition may be involved in the β-lactolin-induced increase of the extracellular dopamine in the hippocampus. It has been reported previously that MAO-B inhibitors enhance cognition in rodents [16,17,18]. In fact, lacto-peptides containing the WY sequence, such as β-lactolin, have been found to improve memory in a scopolamine-induced mouse model of amnesia, in addition to healthy young-adult and aged mice [7]. Such memory-enhancing effects may also result from the increased dopamine level due to MAO-B inhibition.

Consistent with the β-lactolin-induced increase of the extracellular dopamine level in the hippocampus, we found that the dopamine D1 receptor in the hippocampus is vital to the spatial memory-enhancing effects of β-lactolin, as measured in both the Y-maze and the NOLT. The effect of WY dipeptide, the core sequence of β-lactolin, was also partially abolished assessed in the Y-maze test [8], thus dopamine D1 receptor in the hippocampus is involved in the function of β-lactolin-related peptides. On the other hand, the present study did not clearly distinguish the involvement of MAO-B and D1 receptor. Further studies such as using MAO-B knockdown mice would be effective in clarifying these points. The present results are consistent with several previous reports suggesting that stimulation of the D1 receptor in the hippocampus augments spatial memory [9,10]. The dopamine D1 receptor in the hippocampus is expressed mainly in the granule cells of the dentate gyrus and CA1 inhibitory interneurons [19,20]. A study has shown that microinjection of SCH23390, a D1-like receptor antagonist, into the dentate gyrus impairs spatial reference memory in the Morris water maze (MWM) [21]. Consequently, the memory-enhancing effect of β-lactolin may be mediated by the dopamine D1 receptor in the dentate gyrus. As dopamine D1-type receptors promote long-term synaptic potentiation (LTP) at the medial perforant path to the cell synapses in the dentate gyrus [22], β-lactolin may also facilitate LTP in the dentate gyrus, thus improving spatial memory.

Conversely, knockdown of the hippocampal dopamine D1 receptor in the hippocampus had no effect on the memory-enhancing effects of β-lactolin for object information, indicating that this peptide improves object recognition memory through a mechanism that is independent from the hippocampal dopamine D1 receptor. Object recognition memory depends on the functions of both the hippocampus and the prefrontal cortex [23,24] and it has been reported to be impaired by inhibition of D1-like receptors in the prefrontal cortex [25]. Consequently, dopamine D1 receptors in other brain regions, such as the prefrontal cortex, may mediate the memory-enhancing effect of β-lactolin for object information, even when there are no D1 receptors in the hippocampus.

Although there is still uncertainty over the distinct role of the hippocampal D1 receptor in spatial and object memory, a study has shown that D1-like receptors are not involved in all hippocampus-dependent functions. For instance, infusion of SCH23390 into the hippocampus eliminates spatial reference memory in the MWM, but not in inhibitory avoidance, a hippocampus-dependent non-spatial memory task [26]. Thus, the hippocampal dopamine D1 receptor may improve memory functions for a specific type of information and/or in a specific behavioral context. Although the dopamine D1 receptor in the dentate gyrus is involved in spatial memory, it may be less involved in object recognition memory, at least under our experimental conditions. The dentate gyrus is vital for pattern separation of overlapping stimuli [27]. As we used objects with distinct shapes for the NORT, pattern separation by the dentate gyrus may not contribute to the behavioral performance.

Intra-hippocampal treatment with D1-like receptor antagonists has been reported to impair spatial reference memory in the MWM [9]. Mice lacking the dopamine D1 receptor also showed spatial working memory impairment in the MWM [28]. In addition to spatial memories, knockout of the dopamine D1 receptor in the dentate gyrus of the hippocampus also impaired fear-conditioning memory [29]. Our preliminary findings from the MWM revealed that D1 knockdown in the hippocampus tends to delay the latency of reaching the target location in the probe trial following a training period of 5 consecutive days (*p* = 0.061 for *t*-test compared with the control miRNA group) (Appendix A). These results may add support to the theory that the dopamine D1 receptor in the hippocampus plays a role in spatial working memory.

β-Lactoglobulin, a major protein of whey, is rich in WY sequences, and digestion of whey effectively releases β-lactolin, along with other peptide containing WY sequences. In addition, we recently identified another novel bioactive peptide from β-casein that enhances memory functions [6]. Milk proteins including whey and casein would be effective source of bioactive peptides. Recently, we conducted a double-blind, randomized controlled clinical trial that showed that consuming β-lactolin-rich whey peptide could enhance cognitive function in healthy adults [30]. The dopamine precursor levodopa has been reported to enhance cognitive performance in elderly people [31,32]. From the findings of the present study, we speculate that involvement of the dopamine D1 receptor in the hippocampus or other brain areas, including the frontal cortex, may underlie the cognitive improvement seen in healthy adults after consumption of whey peptides rich in β-lactolin. Daily consumption of whey peptides containing β-lactolin and other bioactive peptides may be an effective approach to prevent dementia and cognitive decline.

## Figures and Tables

**Figure 1 nutrients-11-02469-f001:**
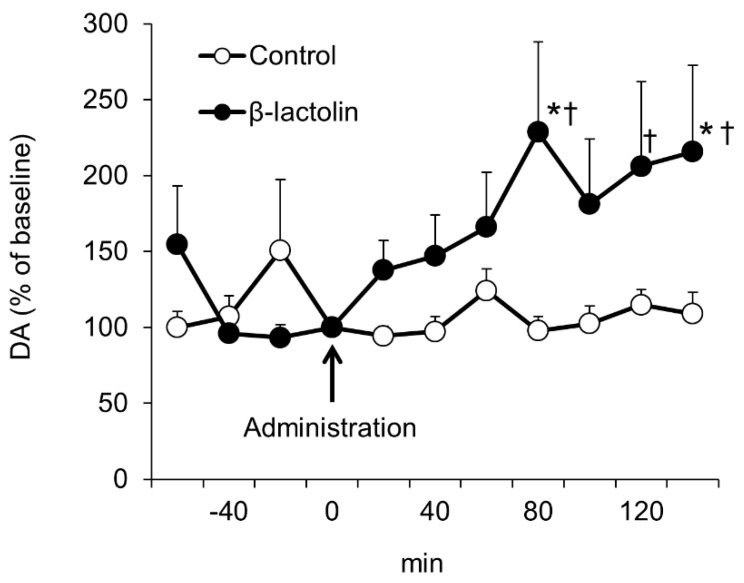
Oral administration of β-lactolin increased the extracellular concentration of dopamine in the hippocampus. Male ICR mice were orally administered β-lactolin (0 or 1 mg/kg), and CSF was collected from the hippocampus via a microdialysis probe, every 20 min between 60 and 140 min post-administration. The DA levels in the collected CSF were quantified using HPLC-ECD. Each value was expressed as the percentage of change from the baseline (DA level at 0 min) of each mouse. Results are presented as the mean ± SEM (*n* = 9–13 mice per group). * *p* < 0.05 versus control (0 mg/kg) group, † *p* < 0.05 versus the baseline of the same group.

**Figure 2 nutrients-11-02469-f002:**
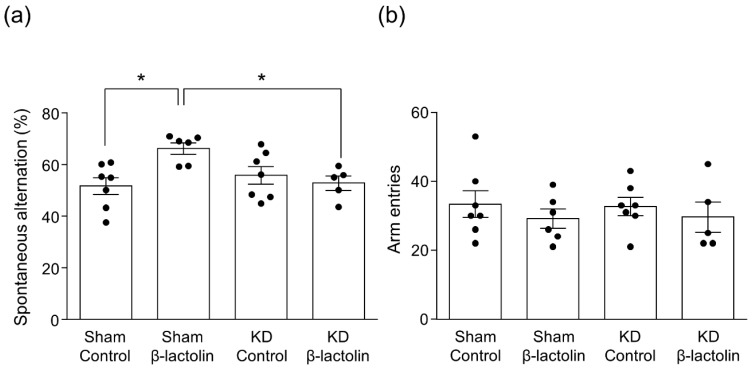
Knockdown of dopamine D1 receptor in the hippocampus abolished the effect of β-lactolin on spatial working memory in the Y maze test. Hippocampal DA D1 receptor knockdown mice or control-knockdown mice were orally administered β-lactolin (0 or 1 mg/kg) and intraperitoneally administered with scopolamine (0.8 mg/kg) after 40 min. Mice were tested on the Y-maze 60 min after oral administration. Spontaneous alternation behavior (**a**) and the number of arm entries (**b**) during the 8 min of the test were measured. Results are presented as the mean ± SEM (*n* = 5–7 mice per group). * *p* < 0.05 versus each group.

**Figure 3 nutrients-11-02469-f003:**
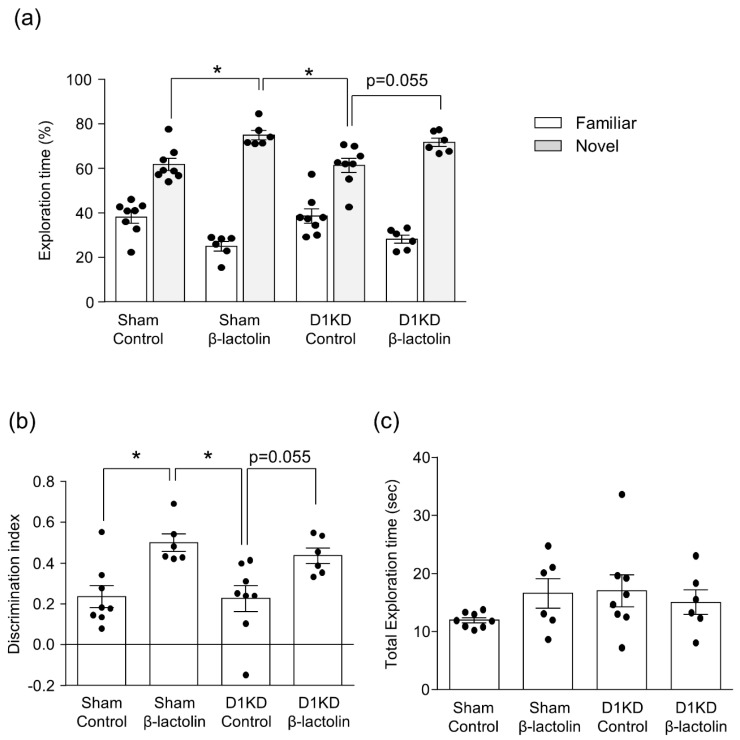
Knockdown of dopamine D1 receptor in the hippocampus spared the memory-enhancing effect β-lactolin in the NORT. Hippocampal dopamine D1 receptor knockdown mice or control-knockdown mice were orally administered β-lactolin (0 or 1 mg/kg) 60 min before the acquisition and recall periods. Time spent exploring the novel and familiar objects during the recall period was measured (**a**). The discrimination index was calculated using the following formula: (novel object exploration time-familiar object exploration time)/(total exploration time) (**b**). The total time spent exploring both objects was calculated (**c**). All values are expressed as the mean ± SEM (*n* = 6–8 mice per group). * *p* < 0.05 versus each group.

**Figure 4 nutrients-11-02469-f004:**
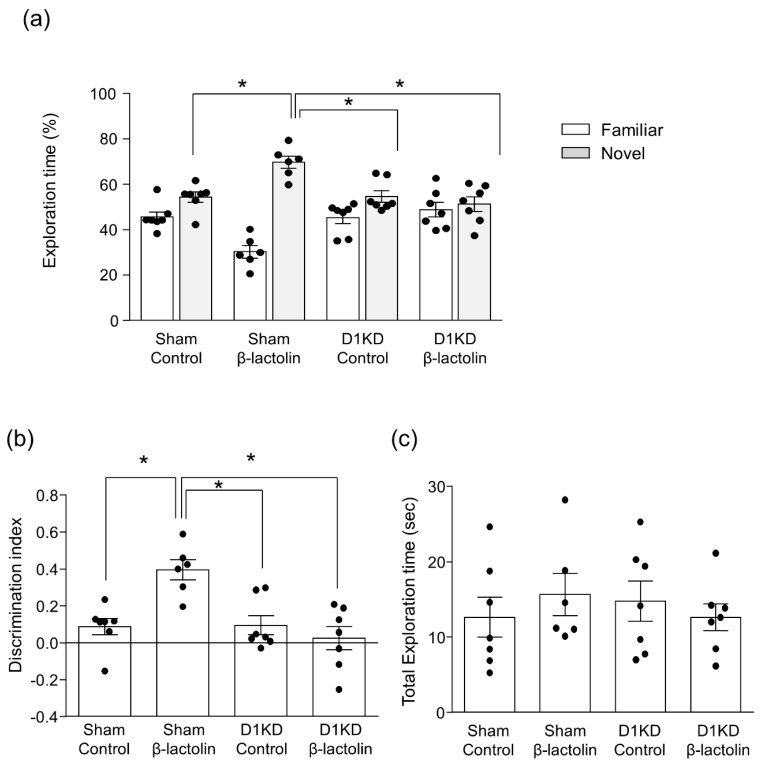
Knockdown of dopamine D1 receptor in the hippocampus abolished the effect of β-lactolin on spatial reference memory in the NOLT. Hippocampal DA D1 receptor knockdown mice or control-knockdown mice were orally administered β-lactolin (0 or 1 mg/kg), 60 min before the acquisition and recall periods. The time spent exploring the novel and familiar locations during the recall period was measured (**a**). The discrimination index was calculated using the following formula: (novel location exploration time - familiar location exploration time)/(total exploration time) (**b**). The total time spent exploring both locations was calculated (**c**). Results are presented as the mean ± SEM (*n* = 6–7 mice per group). * *p* < 0.05 versus each group.

**Table 1 nutrients-11-02469-t001:** Primer list.

Gene	Forward Primer Sequence (5′-3′)	Reverse Primer Sequence (5′-3′)
*Gapdh*	CATCACTGCCACCCAGAAGACTG	ATGCCAGTGAGCTTCCCGTTCAG
*Drd1*	AGATGACTCCGAAGGCAGCCTT	GCCATGTAGGTTTTGCCTTGTGC
*Drd2*	CCTGTCCTTCACCATCTCTTGC	TAGACCAGCAGGGTGACGATGA
*Drd3*	ACCCTGGATGTCATGATGTG	GGCATGACCACTGCTGTGTA
*Drd4*	CCTCTCTTTGTCTACTCCGAGGT	GCCATGAGCGTGTCACAG
*Drd5*	TCCTGGTGTGCTTATGCTTTC	TCAGCTAAGAATCGTTTGGTTTC

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
