# Peer review of "The Lacto-Tetrapeptide Gly–Thr–Trp–Tyr, β-Lactolin, Improves Spatial Memory Functions via Dopamine Release and D1 Receptor Activation in the Hippocampus"

_nutrients, 2019, doi:10.3390/nu11102469_

Round 1

Reviewer 1 Report

The manuscript is presented in logical form and is easy to follow. There are some minor grammatical and punctuation style issues that cause this reader to hesitate in a few places, but overall it is very readable and interesting. It is fairly polished, but might benefit from the eyes of a few more readers first.

I'm trying to think--is there a good way to control for the effects of the peptide on MAO-B to distinguish this possibility from that via the dopamine D1 receptor? Perhaps through the use of rasagiline or some other inhibitor to see if the peptide competes with it?

Also, I'd be very curious to know if they glycine and threonine are essential components of the peptide gaining access to the brain from the diet and/or on its action toward D1 and MAO-B Have you considered trying just the dipeptide WY containing the aromatic side-chains for an effect? Does the tetrapeptide inhibit MAO-B using in-vitro enzymatic assays?

I suspect you can't address all of them, but I have many ideas for follow up experiments.

Author Response

The manuscript is presented in logical form and is easy to follow. There are some minor grammatical and punctuation style issues that cause this reader to hesitate in a few places, but overall it is very readable and interesting. It is fairly polished, but might benefit from the eyes of a few more readers first.

Response: We appreciate your careful reading and the beneficial comments. Now we have corrected the grammars using English editing service.

I'm trying to think--is there a good way to control for the effects of the peptide on MAO-B to distinguish this possibility from that via the dopamine D1 receptor? Perhaps through the use of rasagiline or some other inhibitor to see if the peptide competes with it?

Response: We assume the mechanism of β-lactolin that inhibit MAO-B activity and then increase the dopamine level, and results in the activation of D1 receptor. But present study did not clearly revealed this point. Studies using potent MAO-B inhibitors as you mentioned, or MAO-B knockdown mice would be effective approaches. We have mentioned about this point in the revised manuscript as a limitation.

Also, I'd be very curious to know if they glycine and threonine are essential components of the peptide gaining access to the brain from the diet and/or on its action toward D1 and MAO-B Have you considered trying just the dipeptide WY containing the aromatic side-chains for an effect? Does the tetrapeptide inhibit MAO-B using in-vitro enzymatic assays?

Response: We consider that glycine and threonine residues are not essential for neither the effect nor absorption of β-lactolin. We have previously shown that GTWY, TWY, WY, TWYS peptides exhibit the same effect on cognitive improvement in Y-maze test (Ano, Ayabe et al., Neurobiol Aging, 2018). Single amino acids of tryptophan and tyrosine, and inverse-sequence dipeptide, YW, did not enhance cognitive function (Ano, Ayabe et al., Nutrients, 2019). We have also investigated the effect of both GTWY and WY peptides on MAO-B activity using in vitro enzymatic assay, and both peptides inhibited the MAO-B activity. (Ano, Ayabe et al., Neurobiol Aging, 2018). Taken together, WY sequence is the core of the function of β-lactolin.

I suspect you can't address all of them, but I have many ideas for follow up experiments

Response: We appreciate your comment. We hope these explanations are satisfactory.

Reviewer 2 Report

This research about the role of lactolin peptides on memory functions but in my opinion authors do not discuss actual results and state of the art integrating other close related informations recently published by them. I'm not into this field in deep but up to some point I can understand that more integrative discussion can be done after having read the two following papers that are not, if I'm not wrong, mentioned in the cited literature.

  Identification of a Novel Peptide from beta-Casein That Enhances Spatial and Object Recognition Memory in Mice Por: Ano, Yasuhisa; Kutsukake, Toshiko; Sasaki, Toshinori; et ál.. JOURNAL OF AGRICULTURAL AND FOOD CHEMISTRY   2019, 67, 8160-8167  

Ano, Y.; Ayabe, T.; Ohya, R.; Kondo, K.; Kitaoka, S.;
Furuyashiki, T. Tryptophan-Tyrosine Dipeptide, the Core Sequence
of beta-Lactolin, Improves Memory by Modulating the Dopamine
System. Nutrients 2019, 11 (2), 348.

Author Response

This research about the role of lactolin peptides on memory functions but in my opinion authors do not discuss actual results and state of the art integrating other close related informations recently published by them. I'm not into this field in deep but up to some point I can understand that more integrative discussion can be done after having read the two following papers that are not, if I'm not wrong, mentioned in the cited literature.

Response: We appreciate your careful reading and the comment. As you have mentioned, we should discuss including these recent publications. Now we have cited these articles and added some discussions. These changes are highlighted in red characters in the revised manuscript. We hope that these revisions are satisfactory.
